# Elective Neck Dissection or Sentinel Lymph Node Biopsy in Early Stage Oral Cavity Cancer Patients: The Dutch Experience

**DOI:** 10.3390/cancers12071783

**Published:** 2020-07-03

**Authors:** Inne J. den Toom, Koos Boeve, Daphne Lobeek, Elisabeth Bloemena, Maarten L. Donswijk, Bart de Keizer, W. Martin C. Klop, C. René Leemans, Stefan M. Willems, Robert P. Takes, Max J.H. Witjes, Remco de Bree

**Affiliations:** 1Department of Head and Neck Surgical Oncology, University Medical Center Utrecht, 3584 CX Utrecht, The Netherlands; i.j.dentoom@umcutrecht.nl; 2Department of Oral and Maxillofacial Surgery, University of Groningen, University Medical Center Groningen, 9713 GZ Groningen, The Netherlands; j.boeve@umcg.nl (K.B.); m.j.h.witjes@umcg.nl (M.J.H.W.); 3Department of Pathology and Medical Biology, University of Groningen, University Medical Center Groningen, 9713 GZ Groningen, The Netherlands; 4Department of Radiology and Nuclear Medicine, Radboud University Medical Center, 6525 GA Nijmegen, The Netherlands; daphne.lobeek@radboudumc.nl; 5Department of Oral and Maxillofacial Surgery/Oral Pathology, Amsterdam UMC/Academic Centre for Dentistry (ACTA), 1081 HV Amsterdam, The Netherlands; e.bloemena@amsterdamumc.nl; 6Department of Pathology, Amsterdam UMC (location VU Medical Center), 1081 HV Amsterdam, The Netherlands; 7Department of Nuclear Medicine, The Netherlands Cancer Institute, 1066 CX Amsterdam, The Netherlands; m.donswijk@nki.nl; 8Department of Radiology and Nuclear Medicine, University Medical Center Utrecht, 3584 CX Utrecht, The Netherlands; b.dekeizer@umcutrecht.nl; 9Department of Head and Neck Surgery, Netherlands Cancer Institute, 1066 CX Amsterdam, The Netherlands; m.klop@nki.nl; 10Department of Otolaryngology-Head and Neck Surgery, Amsterdam UMC (location VU University Medical Center), 1081 HV Amsterdam, The Netherlands; cr.leemans@amsterdamumc.nl; 11Department of Pathology, University Medical Center Utrecht, 3584 CX Utrecht, The Netherlands; s.m.willems-4@umcutrecht.nl; 12Department of Pathology, University Medical Center Groningen, 9713 GZ Groningen, The Netherlands; 13Department of Otolaryngology-Head and Neck Surgery, Radboud University Medical Center, 6525 GA Nijmegen, The Netherlands; robert.takes@radboudumc.nl

**Keywords:** oral cancer, lymph node metastases, sentinel lymph node biopsy, elective neck dissection, lymphatics

## Abstract

Background: Sentinel lymph node biopsy (SLNB) has been introduced as a diagnostic staging modality for detection of occult metastases in patients with early stage oral cancer. Comparisons regarding accuracy to the routinely used elective neck dissection (END) are lacking in literature. Methods: A retrospective, multicenter cohort study included 390 patients staged by END and 488 by SLNB. Results: The overall sensitivity (84% vs. 81%, *p* = 0.612) and negative predictive value (NPV) (93%, *p* = 1.000) were comparable between END and SLNB patients. The END cohort contained more pT2 tumours (51%) compared to the SLNB cohort (23%) (*p* < 0.001). No differences were found for sensitivity and NPV between SLNB and END divided by pT stage. In floor-of-mouth (FOM) tumours, SLNB had a lower sensitivity (63% vs. 92%, *p* = 0.006) and NPV (90% vs. 97%, *p* = 0.057) compared to END. Higher disease-specific survival (DSS) rates were found for pT1 SLNB patients compared to pT1 END patients (96% vs. 90%, *p* = 0.048). Conclusion: In the absence of randomized clinical trials, this study provides the highest available evidence that, in oral cancer, SLNB is as accurate as END in detecting occult lymph node metastases, except for floor-of-mouth tumours.

## 1. Introduction

In early stage (cT1-2N0) oral squamous cell carcinoma (OSCC), 20–30% of the patients are diagnosed with occult metastases despite advances in imaging modalities [1,2,3,4]. Conventionally, early stage OSCC patients underwent either watchful waiting or an elective neck dissection (END) for neck staging [5,6]. A recent study showed favorable survival rates for patients who underwent END compared to watchful waiting irrespective of the infiltration depth [7]. However, in this study, the incidence of occult lymph node metastases was up to 45%, suggesting a less accurate diagnostic workup and a different target group, which may hamper generalizability of their results [8]. In the last decade, sentinel lymph node biopsy (SLNB) was introduced in OSCC as a less invasive alternative with lower morbidity rates compared to an END [9]. In early stage OSCC, the most recent meta-analysis reported a pooled sensitivity of 87% and a pooled negative predictive value of 94% for the SLNB procedure in detecting occult metastasis [10].

Because of the low invasiveness and high accuracy rates, the SLNB procedure is implemented in many national head and neck guidelines. In the Netherlands, head and neck oncology care is centralized and the majority of these head and neck cancer institutions use SLNB for staging the clinically node-negative neck in early stage OSCC patients [11,12,13]. Despite the implementation in guidelines, the END is still the staging strategy in early stage OSCC in the majority of medical centers worldwide, even in developed countries with well-organized health care [14]. Cramer et al. showed that SLNB is rarely utilized (<5%) in suitable cases in the United States [15].

Both the END and SLNB have clinically relevant limitations. A pivotal disadvantage of an END is the overtreatment of 70 to 80% of the patients with a more invasive procedure [16]. Several studies reported the differences in complication rates, postoperative morbidity and cost-effectiveness in favor of the SLNB compared to the END procedures [17,18,19,20,21,22]. Conversely, a significantly lower accuracy for SLNB in floor-of-mouth OSCC is reported and is caused by the “shine through phenomenon” [11,12,23,24,25]. This refers to a situation in which the sentinel node is not identified when it is within the flare of radiation from the primary tumour site (Figure 1). The ultimately necessary comparison, e.g., a randomized clinical trial, between the staging accuracy of END versus SLNB is currently not available.

This study presents the accuracy in staging of the cN0 neck and survival rates of both END and SLNB in two large retrospective cohorts. Patients of both cohorts are collected in the same dedicated head and neck centers.

## 2. Results

In total, 390 (44%) END patients and 488 (56%) SLNB patients were used for analysis. Patient and tumour characteristics are given in Table 1. The END cohort contained a higher rate of pT2 tumours (51%) compared to the SLNB cohort (23%) (*p* < 0.001). The anatomical locations differed between the two groups, with more floor-of-mouth (FOM) tumours (34% vs. 27%) and less tongue tumours (50% vs. 62%) in the END cohort (*p* = 0.007). END-treated patients were significantly more often treated with postoperative radiotherapy (34% vs. 11%, *p* < 0.001).

The overall sensitivity of detecting occult metastasis was comparable between the END and SLNB patients (84% vs. 81%, *p* = 0.612). Both groups had similar negative predictive values (NPVs) (93%, *p* = 1.000) (Table 2). Because of dissimilarity in pT staging, we separately analysed the accuracy for pT1 and pT2. In the SLNB cohort, a trend towards a higher though not significantly different sensitivity was observed for pT2 tumours compared to pT1 tumours (88% vs. 76%, *p* = 0.075). In the END cohort, pT2 tumours also showed a higher sensitivity in comparison to pT1 tumours (90% vs. 70%, *p* = 0.010). NPVs did not differ significantly regarding pT stage within the groups. No significant differences were found for sensitivity and NPV between the SLNB and END when corrected for pT stage (Table 2).

### 2.1. Floor-of-Mouth Tumours

In total, 131 (27%) of the SLNB and 133 (34%) of the END patients had tumours located in the FOM. SLNB had a lower sensitivity (63% vs. 92%, *p* = 0.006) and NPV (90% vs. 97%, *p* = 0.057) compared to END. The SLNB had a higher (but not significantly) sensitivity (86% vs. 80%, *p* = 0.315) and NPV (95% vs. 92%, *p* = 0.250) compared to END for other (non-FOM) anatomical locations. When comparing FOM tumours with other non-FOM locations within the SLNB group, there was a lower sensitivity (63% vs. 86%, *p* = 0.008) and NPV (90% vs. 95%, p = 0.113). In contrast, within the END group, at most, a trend towards a higher sensitivity (92% vs. 80%, *p* = 0.114) and a higher NPV (97% vs. 92%, *p* = 0.130) was observed for FOM tumours compared to other anatomical locations. Of the 11 FOM patients with a false negative SLNB, 64% (7/11) had a regional recurrence in level I. In 3 FOM END patients, 1 patient (33%) had a regional recurrence in level I, the remaining patients had a regional recurrence in level II or higher. 

### 2.2. Five-Year Disease-Specific Survival 

The DSS was significantly longer for SLNB pT1 patients (96%) compared to END pT1 (90%, *p* = 0.008), SLNB pT2 (90%, *p* = 0.001) and END pT2 (86%, *p* < 0.001) patients. No significant differences in DSS were seen between the other groups. After the Bonferroni correction, the SLNB pT1 had still a significant longer DSS compared to the other groups: END pT1 (*p* = 0.048), SLNB pT2 (*p* = 0.006) and END pT2 (*p* < 0.001) (Figure 2A).

We furthermore analysed the differences between DSS of the END and SLNB groups divided by anatomical location (FOM vs. other locations, Figure 2B). SLNB-staged patients with a FOM tumour had a longer DSS compared to END FOM patients (98% vs. 87%, *p* = 0.021). The other (non-FOM) SLNB patients had longer DSSs compared to the other END patients (93% vs. 88%, *p* = 0.046). Only the difference between SLNB FOM compared to others END patients remains significant after the Bonferroni correction (*p* = 0.017).

In the DSS analysis with the END and SLNB groups divided for true positives (TP), true negatives (TN) and false negatives (FN) (Figure 2C), the SLNB TN had the longest DSS (97%) and was only not significantly different compared to the END TN (95%). END TN and SLNB TN had significantly longer DSSs compared to the other groups (Figure 2C). Regarding the FN cases, the END FN (23%) had the shortest DSS and was significant different compared to all other groups. The SLNB FN (85%) had a comparable DSS compared to the SLNB TP (86%). After the Bonferroni correction, the differences between SLNB FN (85%) and END TN (95%) and between SLNB TP (86%) with END TN (95%) were not significant anymore. The DSS analysis for SLNB-staged patients with a FOM tumour and divided by TP, TN and FN (Figure 2D) showed the shortest DSS for SLNB FN patients (91%); however, no significant differences were found between these three groups.

### 2.3. Five-Year Regional Recurrence Free Survival

The regional Recurrence Free Survival (RRFS) of END and SLNB patients divided by T stage showed no significant differences between both groups (Figure 2E). However, when comparing the RRFS for the END and SLNB groups divided by anatomical location (FOM vs. other anatomical locations), a significantly shorter RRFS was observed for the SLNB FOM group as compared to the END FOM group (88% and 97%, *p* = 0.008, (*p* = 0.048 Bonferroni corrected) Figure 2F).

### 2.4. Additional Metastasis in the MRND After a Positive SLNB

A histopathologically positive SLNB was found in 107 of the 488 patients (22%). Isolated tumour cells (ITCs) were detected as the largest metastatic deposit in SLNB in 15 patients (14%), micrometastasis (MiM) was detected in 31 patients (29%) and macrometastasis (MaM) was detected in 61 patients (57%). A positive SLNB was followed by an additional (selective) neck dissection in 86% of the patients (92/107). Five patients refused any additional treatment (1 × MiM and 4 × MaM), and the remaining 10 patients received additional radiotherapy instead of a neck dissection. Radiotherapy was required for primary tumour control in 5 cases (2 × ITC, 1 × MiM and 2 × MaM) and was therefore extended to the neck. In 4 cases (1 × MiM and 3 × MaM), radiotherapy was initiated because of 2 or more positive sentinel lymph nodes (SLNs) or extra-nodal extension, and in 1 patient (MiM), extensive surgery was not considered feasible due to major comorbidities. None of these 10 patients developed regional recurrences during follow-up.

Additional non-SLN metastases were found in the neck dissection specimen in 21% of the patients (19/92) with a therapeutic neck dissection after SNLB. Of these patients, the majority (17/19, 90%) had MaM in the SLN as a largest tumour deposit. Patients had a very low risk of having additional metastases if the SLN contained ITCs (8%, 1/13) or MiM (4%, 1/27) compared to MaM (33%, 17/52) (Table 3, *p* = 0.005). 

Disease-specific survival per size of metastasis (ITC, MiM and MaM) did not reach significance, although a trend between ITC and MaM was observed (*p* = 0.091, Bonferonni corrected *p* = 0.182).

## 3. Discussion

This study evaluated the diagnostic value of END and SLNB for early stage OSCC in two large cohorts in the Netherlands. We found an overall sensitivity of 84% in the END cohort and 81% in the SLNB cohort with an NPV of 93% for both cohorts. The SLNB procedure performed worse only in floor-of-mouth (FOM) tumours, while no significant differences between the two cohorts were found for all other anatomical locations. To date, this is the first study that compares END and SLNB procedures for cN0 neck staging in a setting with a numerously powered homogeneous group of patients.

The END cohort showed an overall sensitivity of 84% and an NPV of 93%, based on 19 patients developing regional metastases without evidence of a local recurrence or second primary tumour, resulting in a regional recurrence rate of 7% in the END group. This number is lower than reported by Ganly et al. and Mizrachi et al., who both presented regional recurrence rates up to 15% during follow-up after a negative END [26,27]. A recent meta-analysis showed 274 regional recurrences in 2577 early stage oral cavity cancer patients treated with END (regional recurrence rate 10.6%) [28].

We reported an overall sensitivity of 81% in the SLNB cohort. A trend towards lower sensitivity rates for SLNB procedures can be observed over the last years [10]. In a meta-analysis of Liu et al., recent publications (2009–2016) showed a sensitivity of 86% compared to 92% for the group of early publications (2000–2008). A possible explanation for this trend could be that SLNBs are currently performed in a “routine” instead of research setting, with the learning curves of new SLNB performing surgeons included [23]. Moreover, all our patients are treated following the principle that, in case of a negative SLNB, a watchful waiting strategy is followed. In previous publications with higher sensitivity rates, a complementary neck dissection was often performed as gold standard (or validation) for the SLNB procedure. It is however well known that watchful waiting reveals very small metastases, where these will be missed in up to 15% with routine histopathological examination of a neck dissection specimen (erroneously classified as a true negative) [29,30].

Another explanation of our overall lower sensitivity in the SLNB cohort is a high number of false negatives in the group of patients with FOM tumours compared to the group of patients with non-FOM tumours (sensitivity 63% vs. 86%, *p* = 0.008). As shortly mentioned before in the introduction, lower accuracy rates of SLNB in FOM tumours have been published previously [11,12,23,24,25]. Regional recurrences were particularly located in level I (64%), supporting the theory of missed SLNs in this level by the shine-through phenomenon. Indirectly, this theory was supported by the data of our END cohort showing an excellent sensitivity of 92% in FOM tumours, with only 1/3 patients (33%) showing a regional recurrence in level I. Our data suggest that SLNB in its current form is not reliable enough to detect occult metastases in FOM tumours due to missed (positive) SLNs in level I. However, as shown in survival analyses, the inferior accuracy of SLNB in FOM tumours did not cause lower disease-specific survival (Figure 2B). This implies that salvage neck dissection was successful in most cases, although it should be noted that this may require more extensive surgery and postoperative radiotherapy.

Two additional techniques are recommended in the SLNB surgical guidelines of the consensus meeting in 2018 to overcome the lower sensitivity of the SLNB in FOM tumours [31]. One technique is a super-selective level I neck dissection as described by Stoeckli et al. [25]. The second option is the use of hybrid tracers with a fluorescence label [32,33]. Additionally, Agrawal et al. reported about the use of Tilmanocept, a novel 99m-Technetium tracer with high sensitivity and NPV for detection of occult metastases in the case of FOM tumours [34]. Since this study confirmed a lower accuracy for the SNLB procedure in FOM tumours, we opt nationally to perform the super-selective level I neck dissection as mentioned above. Our study showed that, in the majority of SLNB-positive patients (57%), MaM was detected as the largest tumour deposit, but undeniably, a considerable number of cases had smaller deposits in the SLN (ITCs or MiM). Of 52 patients with MaM treated with a complementary neck dissection, 33% revealed additional (non-SLN) metastases. This number is in line with published literature [35]. As previously published, in only a small number of cases with ITC and MiM, additional non-SLN metastases were found in complementary neck dissection specimens [35]. Our study demonstrated, in only 2 cases with ITC or MiM, additional non SLN metastases (5%). Due to the limited number of studies in the literature, it is difficult to determine the value of complementary neck dissections in the cases of ITCs or MiM. In addition, one has to consider that complementary neck dissection specimens are not examined as meticulously as SLNs. Currently, data is lacking to safely omit therapeutic neck dissections in the cases of ITC or MiM.

Disease-specific survival of the false-negative cases of both cohorts addresses an important finding. SLNB FN patients had an almost equal DSS compared to the true-positive patients. In contrast, in the END FN cohort, a dramatic decreased survival was shown compared to the END TP patients. Our data clearly underlines the importance of correct staging using a minimally invasive method given this inferior survival for FN patients, specifically in the END cohort.

One of the limitations of this study remains its retrospective design. In the END cohort, a considerable number of the patients were diagnosed as clinically N0 based on potentially dated ultrasound, computed tomography (CT) and magnetic resonance imaging (MRI) scanners and before widespread application of 2-deoxy-2-(18F)fluoro-D-glucose positron emission tomography (FDG-PET) for staging purposes. There is also a significant difference in pT stage with more pT2-staged tumours in the END cohort. A possible explanation is that the majority of patients in the END cohort was selected based on depth of invasion (>4 mm), inevitably resulting in less pT1 tumours. For that reason, we compared also the sensitivity and NPV between SLNB and END divided by pT stage and found no significant difference. Besides the diameter of the tumour reflected in pT stage in the 7th TNM classification, nowadays, the 8th edition is used [36,37,38]. In the 8th edition, depth of invasion is newly incorporated for T stage, and therefore, our results could not directly be translated to the 8th TNM classification [39,40,41]. Another important difference between both groups is the prolonged follow-up in the END cohort. Although we expected (and identified) regional recurrences particularly in the first 2 years, a longer follow-up could result in more regional recurrences and/or disease-specific deaths.

## 4. Materials and Methods 

Patients were included from five Dutch head and neck centers that are now all performing SLNB as the standard diagnostic modality for early stage OSCC. Patients were included if they were treated between 1990 and 2015 for the END cohort and between 2007 and 2018 for the SLNB cohort. The overlap in years of treatment (2007–2015) between END and SLNB is caused by the different times of introduction of the SLNB procedure in individual hospitals involved in this study. To keep the cohorts as homogeneous as possible, END patients were collected in the same centers and were treated before these centers introduced the SLNB. Consequently, the END cohort provided longer follow-up with respectively a median follow-up of 4.5 (interquartile range (IQR) 2.5 to 7.3) years versus 2.2 (IQR 1.0 to 4.1) years (*p* < 0.001). Tumours are classified according to the 7th TNM classification system. All clinical and histopathological data were retrospectively collected from the electronic patient files. Information regarding SLN location and pathological LN assessment in both END and SLNB are listed prospectively and standardised as part of the daily clinical practice in the participating centers. All patients were staged as cN0 by palpation and imaging (ultrasound, CT and/or MRI). In the case of suspected lymph nodes, ultrasound-guided fine needle aspiration cytology was performed. 

The SLNB procedure was performed according to the European Association of Nuclear Medicine/Sentinel European Node Trial joint practice guidelines and has been described extensively before by some of the participating institutions [11,12,42]. The procedure corresponds with the recommendations in the guidelines from the 2018 SLNB consensus conference held in London [31,43]. Shortly, the procedure consists at least of three modalities: preoperative visualization of SLNs (using peritumoural injections of (^99m^Tc)Tc-nanocolloid) with planar dynamic and static lymphoscintigraphy, including single photon emission computed tomography and computed tomography (SPECT/CT) scanning, in a one or two day protocol; intraoperative detection and extirpation of SLNs performed using a handheld gamma probe; and harvested SLNs assessed using step-serial sectioning with haematoxylin and eosin staining and additional keratin staining [42]. Metastasis were classified as isolated tumour cells (ITC) (size ≤ 0.2 mm), micrometastasis (MiM) (size > 0.2 mm and ≤ 2 mm) or macrometastasis (MaM) (size > 2 mm) according to Hermanek et al. [44]. SLNB-negative patients had a watchful waiting strategy as reference, while SLNB-positive patients were treated by neck dissection and/or radiotherapy on the neck. In the case of a negative SLNB, patients followed a strict follow-up regimen; patients visited our outpatient clinic in the first year every 2 months, in the second year every 3 months, in the third year every 4 months, and in the fourth and fifth years every 6 months.

The END cohort consisted of early stage OSCC patients with a (selective (level I-III/IV) or modified radical) neck dissection as part of the primary treatment. Data of these END patients were available in 4 of the participating centers. In these centers, END was the first choice over watchful waiting when tumour depth of invasion was estimated as above 4 mm [5,6,45]. In 2 centers, frozen sections of one or two clinically most suspicious lymph nodes were routinely assessed intraoperatively during END, and in the case of a detected metastasis, the END was converted to a therapeutic modified radical neck dissection (MRND). In the case of negative fresh frozen material, only level I–III ENDs were performed with postoperative lymph node assessment using conventional haematoxylin and eosin staining on formalin-fixed paraffin-embedded tissue. 

### 4.1. Ethical Consideration

The study was conducted in accordance with the Declaration of Helsinki, and the protocol was approved by the Ethics Committee of UMC Utrecht (no. 17/766). The Internal Review Board waived the requirement for the investigator to obtain signed consent forms for all subjects. All samples and data were handled according to the General Data Protection Regulation.

### 4.2. Statistical Analysis

Categorical data are given with N and percentage using the Fisher’s exact test and Chi-square test to analyse differences. The Mann–Whitney U (skewed distribution) and Student’s t-test (normally distributed) were used for continuous data and are given with respectively the median and interquartile range (IQR) and the mean with the standard deviations (SD). True positives (TP), true negatives (TN) and false negatives (FN) are defined as respectively pN+ (TP), pN0 without (TN) and pN0 with (FN) detection of a regional recurrence after primary treatment. The Kaplan–Meier method and log-rank test was performed for survival analysis (disease-specific survival (DSS) and regional recurrence free survival (RRFS)). Five-year DSS was defined as the time from treatment till death or last clinical visit. Deaths caused by the early stage OSCC within five years after treatment were counted as an event. Five-year RRFS was defined as the time from treatment till regional recurrence without local disease or last clinical visit. Regional recurrences without local disease and within five years after treatment were counted as events for the RRFS. Regional recurrences with presence of local recurrence or second primary tumours were excluded. A Bonferroni correction was used for log-rank tests for multiple testing and was defined by multiplying each *p*-value by the total number of comparisons. A value of *p* < 0.05 was considered statistically significant. All statistical tests were analysed using IBM SPSS Statistics 23 for Windows (Statistical Package for the Social Sciences, Inc., Chicago, IL, USA).

## 5. Conclusions

In conclusion, detection of lymph node metastases in oral cancer using sentinel lymph node biopsy is as accurate as elective neck dissection, except for floor-of-mouth tumours. SLNB showed higher disease-specific survival rates as compared to the END, also after stratifying for different pT stages or anatomical locations. As randomized controlled trials comparing the accuracy of SLNB with the routinely used END are currently lacking, this retrospective cohort study provides the highest evidence of the effectiveness of SLNB in oral cancer.

## Figures and Tables

**Figure 1 cancers-12-01783-f001:**
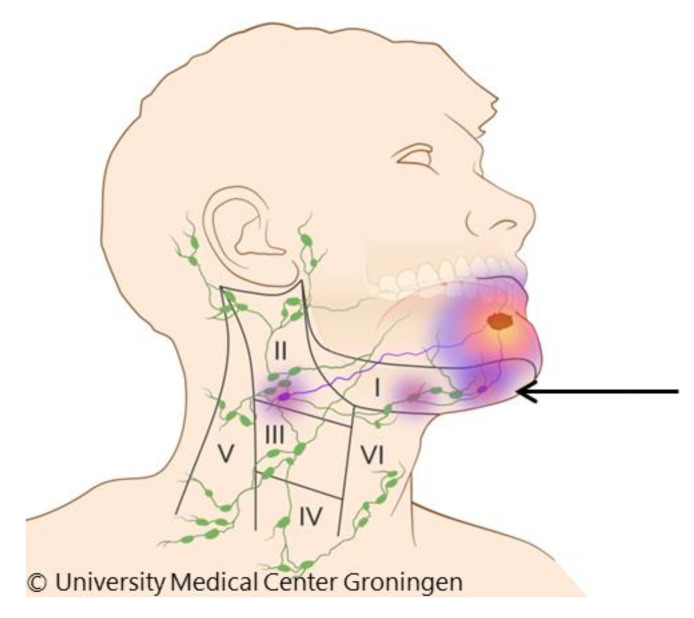
Shine through phenomenon: Radiation flare of the primary tumour overshines the hotspot of a sentinel lymph node in close proximity to the primary tumour (arrow).

**Figure 2 cancers-12-01783-f002:**
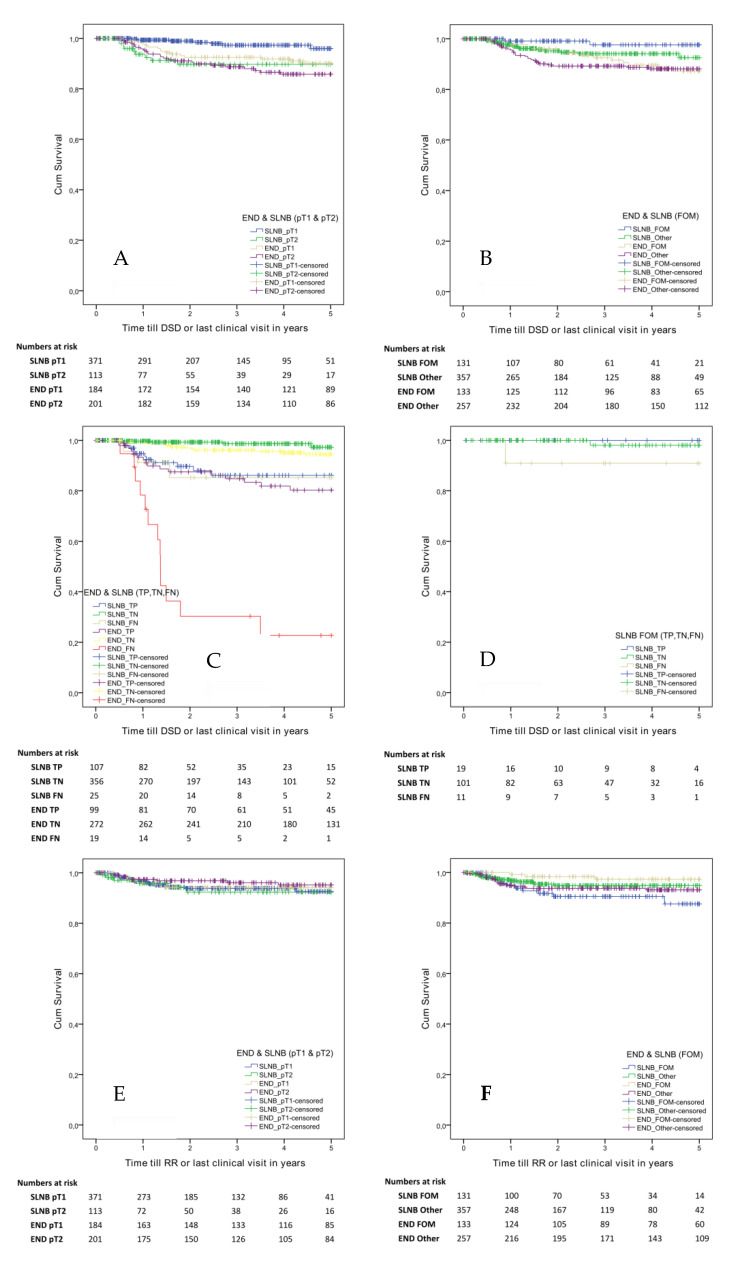
Survival analyses: Disease-specific survival between END and SLNB patients divided by T stage (**A**) and by anatomical location (**B**). Because of the low number of pT3 (*n* = 8) and pT4 (*n* = 2)-staged patients for each of the staging methods, these pT stages were excluded from the disease-specific survival analysis divided by T stage (**A**). Disease-specific survival analysis of the END and SLNB groups divided for true positives, true negatives and false negative patients (**C**). SLNB-staged patients with an FOM tumour were also divided by true positive, true negative and false negative patients (**D**). Regional recurrence free survival between END and SLNB patients divided by T stage (**E**) and by anatomical location (**F**). Abbreviations: END, elective neck dissection; SLNB, sentinel lymph node biopsy; FOM, floor of mouth.

**Table 1 cancers-12-01783-t001:** Patient and tumour characteristics.

Characteristic	Category	SLNB		END		
		*n*	(%)	*n*	(%)	*p*-Value
Total		488	(56)	390	(44)	
Gender	Female	237	(49)	178	(46)	0.377
	Male	250	(51)	212	(54)	
Age at treatment (years)	Median (IQR)	63	(55 to 69)	62	(53 to 70)	0.767
	Range	20 to 93		22 to 95		
cT (7th)	T1	335	(69)	136	(35)	<0.001
	T2	153	(31)	254	(65)	
pT (7th)	T1	371	(76)	184	(47)	<0.001
	T2	113	(23)	201	(51)	
	T3	4	(1)	3	(1)	
	T4	0	(0)	2	(1)	
pN	Negative	381	(78)	291	(75)	0.264
	Positive	107	(22)	99	(25)	
Metastasis size	ITC (<0.2 mm)	15				
	Micro (0.2 to 2.0 mm)	31				
	Macro (>2 mm)	61				
Postoperative RTx	Yes	52	(11)	131	(34)	<0.001
	No	436	(89)	259	(66)	
Location	Tongue	302	(62)	196	(50)	0.007
	FOM	131	(27)	133	(34)	
	Cheek/Buccal/Trigonum	34	(7)	35	(9)	
	Others	21	(4)	26	(7)	
END levels	I to III	NA		300	(77)	NA
	I to IV	NA		16	(4)	
	I to V	NA		74	(19)	
Follow-up	Time in years median (IQR)	2.2	(1.0 to 4.1)	4.5	(2.5 to 7.3)	<0.001
	Range (years)	0.0 to 9.7		0.0 to 20.8		
	Regional recurrences	25	(5)	19	(5)	1.000
	Deceased	52	(11)	140	(36)	<0.001
	Deceased by disease	18	(4)	45	(11)	<0.001

Abbreviations: END, elective neck dissection; SLNB, sentinel lymph node biopsy; IQR, interquartile range; ITC, isolated tumour cells; MiM, micrometastasis; MaM, macrometastasis; FOM, floor of mouth; RTx, radiotherapy.

**Table 2 cancers-12-01783-t002:** Sensitivity and negative predictive value.

(Sub)Group		SLNB		END		
		%	(95% CI)	%	(95% CI)	*p*-Value
Overall	Sensitivity	81	(73 to 87)	84	(76 to 90)	0.612
	NPV	93	(91 to 95)	93	(90 to 95)	1.000
pT1 *	Sensitivity	76	(65 to 85)	70	(51 to 84)	0.637
	NPV	94	(91 to 96)	94	(90 to 96)	1.000
pT2 *	Sensitivity	88	(77 to 96)	90	(82 to 96)	0.776
	NPV	91	(83 to 96)	94	(88 to 97)	0.565
FOM **	Sensitivity	63	(44 to 80)	92	(79 to 98)	0.006
	NPV	90	(85 to 94)	97	(91 to 99)	0.057
Other locations **	Sensitivity	86	(78 to 92)	80	(70 to 88)	0.315
	NPV	95	(92 to 97)	92	(88 to 95)	0.250

Abbreviations: END, elective neck dissection; SLNB, sentinel lymph node biopsy; FOM, floor of mouth; NPV, negative predictive value. * pT1 versus pT2 within the SLNB group: sensitivity *p* = 0.075 and NPV *p* = 0.415. * pT1 versus pT2 within the END group: sensitivity *p* = 0.010 and NPV *p* = 1.000. ** FOM versus other locations within the SLNB group: sensitivity *p* = 0.008 and NPV *p* = 0.113. ** FOM versus other locations within the END group: sensitivity *p* = 0.114 and NPV *p* = 0.130.

**Table 3 cancers-12-01783-t003:** Additional positive lymph nodes in complementary neck dissection specimen in the cases of positive SLNB.

		Yes	No	Other	Total
SLNB	ITC	1	12	2	15
	MiM	1	26	4	31
	MaM	17	35	9	61
Total		19	73	15	107

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
