# Peer review of "Elective Neck Dissection or Sentinel Lymph Node Biopsy in Early Stage Oral Cavity Cancer Patients: The Dutch Experience"

_cancers, 2020, doi:10.3390/cancers12071783_

Round 1
Reviewer 1 Report
The manuscript has elaborated the methodology and rationale and reflects the good design of the study.
Has the study used any IHC marker to visualize the ITCs (Individual Tumour Cells) or has it relied on evaluation of the same by H&E alone? In that case, there’s a very high chance of ITCs being underestimated.
The focus on the limitations (described from line 242) serves to reinforce the idea that substantial amounts of previous data and studies (based on dated or now redundant guidelines/technology/equipment) may actually show a ‘stage migration’ with reassignment of TNM stage to a higher one. Another worry is the amount of data that has originated from developing countries that have a high burden of OSCC but have only relied on ‘essential TNM staging.’
The abbreviation NPV appears first in line 99 but is expanded only in line 108.
Lines 249 and 251 introduce a term ‘T stadium’ that is not commonly used.
Reviewer 2 Report
Thank you for the opportunity to review the manuscript titled ELDN or SNB in early stage oral cavity cancer. The study compares a SNB guided neck management with a historical control of END patients with early stage oral cavity cancer managed in the Netherlands. The study reports being the only study comparing the two approaches recognizing that it is not a clinical trial. The question addressed is important and does add to the literature. The study is approriately designed and well written and of importance in order to see the field move toward less invasive procedures
I have a few comments/suggestions for the authors
- The authros mention being the only study comparing the two. There is a national cancer database study by Cramer et al. In the discussion ity would be worthwhile considering discussing the results and also the strengths of the current study compared to a NCDB study
- In the methods section
- Data was obtained from different Dutch head and neck centers. Was a retrospective review performed at each center?. I recall that the Dutch system have excellent registries and if so would highlight that. I apologize if I may have missed where it mentioned this.
- There are overlapping years for END and SNB – however the authors note that they tried to include ND collected in the same centers as SNB prior to the introduction of SNB. I feel a bit more specfic detail is warranted to see if at he centers there were still overlapping cases ie of of END performed when SNB was already introduced. Themost important aspect of this study is managing the selection bias of ND over SNB and if any END were performed during the time of SNB what would the reason be. Maybe a supplementary table. The concern is the fact that in the ND group there were more patients undergoing EBRT post op. I thnk the discussion would benefit for a more indepth discussion about this – why it would be the case and how a potential selection bias exists, at the same time would this overestimate the benefit of ND
- After introduction of SNB at the respective centers could the authors calrify if SNB were routinely performed on all cT1 T2 N0 patients and if not what are the deciding factors.
- In the discussion under limitations – please include a senteance about differences in follow up time which may account for some of the differences in survival analysis
- Since the 7th edition was used would current T3 (ie > 10 mm DOI) have been included in the study and would all have received a SNB or would some of these have received a neck dissection. Was there a difference in the proportion of patients in ND group based on path staging that would fall in T3 category currently
- In the results the DSS were longer in SNB – while I would expect lower morbidity with SNB ehy would the authors think there would be improved survival as we wouldnto expect END in of itself to increase mortality – is there an underlying bias or confounder. Similalry the RRFS the same but I would have expected that even with successful salvage the Regional recurrence free survival would be higher if there were more regional recurrence as the event for analysis would be regional recurrence or survival? This would be independent of the overall survival after salvage of recurrence.
- In the discussion the authors mention the groups were homogenous but are they truly homogenous groups especially if the the ND had more EBRT
- The authors mention the studyby Ganly- which showed failure in the contralateral neck . In your study did you collect the number of cases where a Unilateral ND would have been in teheory performed based on the disease location but the SNB identified a contralateral node and whether these contralateral nodes were SNB positive
- For the SNB negative patient would you be able to put in your protocol for watchful waiting in terms of follow up and imaging – inparticualr are all patients surveilled at the same hospital with the same imaging department and how do you habdle patienst travelling from long distances.
- Lastly the discussion would benefit from a comment of your current approach to neck management of FOM cancers. You mention the options but would be good to get your current approach
Reviewer 3 Report
Elective Neck Dissection or Sentinel Lymph Node Biopsy in Early Stage Oral Cavity Cancer Patients: The Dutch Experience. Authors: Inne J. den Toom et al. The work presented is an interesting investigation that contributes novel aspects to the design of the treatment and prognostic prediction of oral cancer. A wide series is presented. In my opinion it is potentially acceptable for publication in Cancers, although I think that some modifications should be made that would improve the final result: INTRODUCTION Authors should better define what is understood by early carcinoma, with a specific bibliographic support for this concept, even reproducing literally, in quotes, what is considered in the literature as early carcinoma. MATERIAL AND METHODS Although a significant number of statistical studies have been carried out to treat the results of the study, in my opinion an explanation of the reason that explains the differences in the mean follow-up of patients undergoing END and patients undergoing SLND would be necessary . Furthermore, I think it would be interesting to make some consideration on the reason because of in this series the involvement of women and men is practically similar (there are no statistical differences between the sexes). Finally, is there any other hypothetical reason that justifies the low sensitivity of SLNB in the floor of the mouth, different from the Shinethrough phenomenon? If so, it should be commented on in the discussion, since it is strange that this very important aspect of the results can only be justified in a single hypothesis. Finally, when talking about tumor thickness, the paper should beg for " Importance of tumor thickness measurement in prognosis of tongue cancer. Gonzalez-Moles MA, Esteban F, Rodriguez-Archilla A, Ruiz-Avila I, Gonzalez-Moles S. Oral Oncol. 2002 Jun; 38 (4): 394-7" should be cited.Author Response
Please see the attachment.
